# The Moderating Role of Emotion Regulation in the Recall of Negative Autobiographical Memories

**DOI:** 10.3390/ijerph18137122

**Published:** 2021-07-02

**Authors:** Desirée Colombo, Silvia Serino, Carlos Suso-Ribera, Javier Fernández-Álvarez, Pietro Cipresso, Azucena García-Palacios, Giuseppe Riva, Cristina Botella

**Affiliations:** 1Department of Basic Psychology, Clinic and Psychobiology, Universitat Jaume I, 12071 Castellón, Spain; azucena@uji.es (A.G.-P.); botella@uji.es (C.B.); 2Humane Technology Lab, Università Cattolica del Sacro Cuore, 20123 Milan, Italy; silvia.serino@unicatt.it (S.S.); javier.fernandezkirszman@unicatt.it (J.F.-Á.); giuseppe.riva@unicatt.it (G.R.); 3Department of Psychology, Università di Torino, 10124 Torino, Italy; p.cipresso@auxologico.it; 4CIBER Fisiopatología Obesidad y Nutrición (CIBERobn), Instituto Salud Carlos III, 28029 Madrid, Spain; 5Applied Technology for Neuro-Psychology Lab, IRCCS Istituto Auxologico Italiano, 20149 Milan, Italy

**Keywords:** emotion regulation, rumination, cognitive reappraisal, ecological momentary assessment, autobiographical memory

## Abstract

When facing a negative event, people implement different strategies to regulate ongoing emotions. Although the previous literature has suggested that the emotional intensity of a negative episode is associated with the characteristics of the subsequent autobiographical memory, it is still unknown whether emotion regulation (ER) moderates this relationship. In the present study, we provided undergraduate students with a smartphone-based diary to report a negative episode immediately after its occurrence and rate the momentary use of two ER strategies: cognitive reappraisal and rumination. To explore autobiographical memory, two “surprise” recall tasks were performed one week and one month after the event. According to the results, cognitive reappraisal was linked with better memory performances, and a tendency to retrospectively underestimate the negativity of highly intense events was observed only in participants adopting high rates of this strategy. Conversely, intense rumination was found to be associated with less detailed memories of emotionally intense events, as well as with higher emotional involvement with negative episodes over time, regardless of their intensity. Together, our results support the maladaptive role of rumination and the adaptive influence of cognitive reappraisal on autobiographical memory.

## 1. Introduction

For a long time, emotion and cognition were considered two independent aspects of the human being [1], but current evidence shows that cognitive and emotional processes are reciprocally determined and interact continuously [2,3]. Autobiographical memories are no exception, and a growing body of literature has revealed the essential role played by emotions in encoding, storing, and recalling past personal events [4,5,6,7]. 

The term ‘autobiographical memory’ refers to people’s set of memories of their own lives [8], which can include both marginal episodes or emotionally intense events [9]. Autobiographical memories are essential to guide future behaviors and decisions, as well as to promote a sense of continuity of the Self and facilitate social interactions [10]. Interestingly, the emotional correlates of an event have been shown to influence the characteristics of the subsequent memory. According to previous studies, highly arousing memories are associated with enhanced retrieval of central details [11] and increased vividness [12]. Furthermore, whereas positive memories are characterized by an enhanced recall of peripheral and sensory details [13,14,15], negative memories have been linked to greater recall of central features [16,17] and increased accuracy [18]. 

Despite ample evidence of the relationship between emotions and autobiographical memories, the specific role of emotion regulation (ER) in shaping memory retrieval processes has been less explored. ER is the ability to implicitly or explicitly modify an emotional state to create adaptive emotional responses and reach desirable goals [19,20]. People may attempt to regulate their emotions at different timepoints in the emotion generation process [21]. This includes before the emotion is generated (i.e., antecedent-focused strategies such as situation selection, situation modification, attentional deployment, or cognitive change) or after the emotion occurs (i.e., response-focused strategies such as response modulation). In this vein, a recent revision of the literature has pointed out the different cognitive, behavioral, and affective correlates of ER strategies [22], which have been traditionally classified as adaptive or maladaptive based on the associated outcomes [23,24,25]. Of these, cognitive reappraisal and rumination are among the most frequently investigated ER strategies in relation to mnemonic processes, being the former an adaptive strategy and the latter a maladaptive one. 

Cognitive reappraisal and rumination are both antecedent-focused strategies that imply cognitive engagement with an emotion-eliciting stimulus. Though, there is evidence showing their different cognitive, behavioral, and affective implications. Cognitive reappraisal refers to the attempt to reconstruct and re-elaborate a stimulus to change its meaning and emotional impact [21]. Because it is cognitively demanding [26], this strategy has been shown to be implemented more often in situations that are less emotionally intense [27]. On the other hand, rumination refers to the process of persistently thinking about one’s feelings and the associated causes and consequences [28]. Unlike reappraisal, this strategy is more likely to be adopted when experiencing intense negative emotions [29,30]. Thus, cognitive reappraisal and rumination are opposite in their nature and entail different consequences. The use of cognitive reappraisal implies an active and explicit attempt to change the interpretation of a stimulus to modify its emotional impact [26]. On the contrary, rumination involves a passive attitude toward the experience, characterized by negative and repetitive thoughts that do not promote a positive reframing of the situation [31]. In this sense, reappraisal has been shown to be effective in reducing negative emotions [32], whereas the use of rumination has been shown to maintain and prolong negative affect [24]. 

Consistent with these findings, these two strategies have been shown to entail different outcomes on memory. The previous literature has suggested that reappraisal promotes a more in-deep analysis of a situation [33], thus being associated with enhanced memory performances and higher recall of details [33,34,35,36]. Reappraisal has been proposed as a strategy that encourages a more positive reframing of a negative situation, thus leading to more positively valenced recalls [37,38]. Differently, rumination has been indicated as a potential mechanism underlying overgeneral memories (i.e., memories that lack details), which might be the consequence of the continuous rehashing and emphasis on the negative emotional correlates of an episode [39,40]. More specifically, ruminative thinking has been hypothesized to lead to focus one’s attention on the general information rather than on specific details, which in turn is likely to result in the retrieval of less specific memories [41,42,43,44].

Interestingly, the previous literature has been mainly based on laboratory studies and retrospective assessments of events [33,34,35,36,37,38], thus making these findings not completely generalizable to real-life situations [45]. First, the role of ER in the recall of a negative stimulus (e.g., a video or a picture) might strongly differ from its role in the recall of a real-life negative episode. Second, the previous studies typically adopted a trait-based conceptualization of ER [46], which was considered a stable and cross-situational tendency of an individual [22]. There is evidence, however, that ER is a situated process shaped by momentary situational and contextual factors, which are difficult to capture through trait-based questionnaires [47,48]. Third, the previous literature assumed that there was a direct association between ER and memory. However, it is also possible that ER influences memory in a more complex manner. Although it is true that the emotional intensity of an event is associated with the qualities of the subsequent memory, ER could affect this relationship so that the intensity of the event is more or less strongly associated with memory qualities depending on the strategy adopted (i.e., moderation). Finally, the use of free recall tasks, which has been the mainstream method to assess the number of details recalled in laboratory settings, seems challenging in the case of autobiographical memories. Applying this methodology would, for instance, require recording an eliciting event in a controlled setting, which could be subsequently compared to the associated recalled memory (see for example [33]). As noted in past research, however, emotionally relevant situations and the associated ER mechanisms are more easily captured in real-life settings [22]. Alternatively, the recalled memory could be compared to the description of the episode provided by the individual at the time of its occurrence, which in turn might be biased [49,50]. In other words, the use of free recall tasks to assess the memory of real-life episodes may not be appropriate, thus making self-report questionnaires more suitable to explore autobiographical memory and its phenomenological characteristics [51]. 

In sum, while the available studies support the hypothesis that different strategies may influence mnemonic processes, the role of ER in the recall of real-life negative events is still unclear. However, understanding this relationship would be of paramount importance. Autobiographical memories represent the core of an individual’s affective architecture to build a coherent Self and a sense of purpose in life [52], and reminiscing about the past is considered an effective strategy to regulate emotions in the present. If it is true that ER affects the encoding of a stimulus and the subsequent recall of the associated memory, the use of adaptive strategies as opposed to maladaptive ones in face of adverse life events could facilitate the reframing of such episodes, promote the construction of more coherent life stories and encourage the use of past memories to regulate emotions in the present. 

### The Current Study

In the present study, we attempted to address some of the aforementioned issues by adopting an ecological and context-dependent perspective. The main aim was to investigate whether ER moderated the relationship between the emotional intensity of a personal negative event and the associated autobiographical memory. To do so, we asked participants to record through a smartphone-based diary one specific negative event, to rate its negative valence, and to report the ER strategies used to deal with the emotions experienced. They were asked to do so in real-time or as close as possible to the event’s occurrence. To disentangle the relationship between ER and memory over time, two “surprise” recall tasks were performed one week and one month after the episode assessing memory phenomenology (i.e., clarity; sensory information; temporal information; emotional involvement; thoughts and feelings) [53] and participants’ subjective appraisal of the event over time.

Based on the previous literature [11,16,17,18,54,55], we expected events with higher emotional intensity to be recalled with greater vividness. Furthermore, we hypothesized that ER would correlate with autobiographical memory characteristics and moderate its association with an event’s emotional intensity. Considering the opposite nature of rumination and cognitive reappraisal, as well as their different cognitive and affective consequences, we hypothesized that each strategy would entail different outcomes on autobiographical memory.

As ruminative thinking entails persistently rehashing an episode, and its causes and consequences, we hypothesized that rumination would moderate the association between the emotional intensity of an episode and one’s emotional involvement with it. Although emotionally intense episodes might be recalled with higher emotional involvement, we expected intense rumination to be associated with heightened emotional involvement, regardless of the emotional intensity of the event. Furthermore, as ruminative thinking has been identified as a potential mechanism underlying overgeneral memories [39], we expected intense rumination to be associated with less detailed memories and to moderate the relation between emotional intensity and memory details. More specifically, episodes with higher emotional intensity have been shown to be discriminated faster and recalled with more details [16,17], which would reflect an evolutionary mechanism that increases the salience of threatening events in terms of survival [56]. Nevertheless, we hypothesized that the use of intense levels of rumination would decrease the clarity of the event over time.Because cognitive reappraisal entails a deep conceptual analysis of a situation to change its meaning [33], we hypothesized that participants with a high use of reappraisal would remember the event in more detail and with greater clarity. In addition, the use of cognitive reappraisal was hypothesized to lead to a higher elaboration of and distancing from the episode [37,38]. Although people have been shown to typically overestimate the negativity of past emotional experiences [57], we hypothesized that reappraisal would moderate the association between the emotional intensity of an event and its appraisal over time. Since cognitive reappraisal is meant to decrease the emotional impact of a negative stimulus [21], we expected that participants with intense reappraisal adoption, but not low-reappraisal use, would show a higher positively valenced appraisal of emotionally intense events over time.

## 2. Materials and Methods

### 2.1. Sample

An a priori power analysis was conducted using G*Power (‘linear multiple regression: fixed model, R2 increase’, Heinrich Heine University Dusseldorf, Dusseldorf, Germany), considering the moderations (i.e., the interaction between ER strategies and emotional intensity to predict autobiographical memory characteristics) as main analyses. Assuming an overall medium effect size of 0.15, a significance level of 5%, a statistical power of 80%, the sample size calculation resulted in a sample of *n* = 55. Nevertheless, a post hoc calculation of the achieved power is also provided in the results section. This study was approved by the Ethical Committee of the Jaume I University (reference: 16/2018). All the participants gave their informed consent to participate.

Taking into account potential dropouts throughout the study, a larger sample was recruited. In total, seventy-two undergraduate students were recruited through online advertisements at Jaume I University (Castellon, Spain). They were asked to report a negative event on a weblink as soon as possible after its occurrence. Events reported with over a 12-h delay were not considered valid and, therefore, were not included in the study. Accordingly, one participant was excluded, thus resulting in a final sample of 71 participants (8 males/63 females). The sample’s mean age was 21.87 years (SD = 4.01), and the average number of years of education was 11.19 (SD = 2.41).

### 2.2. Measures

#### 2.2.1. Recording the Negative Event

Participants were provided with the web link to a survey, which could be accessed with smartphones or laptops. The instructions were as follows: “In the coming days, if a significant negative event happens in your life, enter this weblink. You will be asked to describe the event that has just occurred and answer a couple of questions. Please, read these instructions carefully. By “specific event”, we are referring to something specific (i.e., an episode) that happened during your day and took place in a specific spatial location and a defined time period. Having an argument with your best friend or losing your wallet on the bus, for example, can be considered specific events. Also, it has to be a “significant event”, i.e., this event has to be significant to you in such a way that it is likely to affect your mood and influence your behaviors and/or thoughts. Please, write the event down at the weblink as soon as possible after its occurrence.”

At the web link, participants were asked to provide the temporal details and a written description of the episode, as well as to rate its negativity using a Visual Analogue Scale (“On a scale from 0 (not unpleasant) to 100 (completely unpleasant), how would you rate the valence of event?”). Participants were also asked to rate the intensity of seven negative emotions on a 5-point Likert scale (“To what extent have you experienced these emotions?”: sadness, anger, fear, anxiety, disgust, guilt, shame). The items to assess negative emotions were selected based on previous research [58] and discussions in our research team. These Likert scales were used to calculate ‘emotional intensity’, a composite score obtained by summing up the ratings of the seven emotions. The internal consistency of this new variable was *α* = 0.71, and significant moderate-to-large correlations were observed between this composite score and the seven negative emotions (sadness: *r* = 0.674, *p* < 0.001; anger: *r* = 0.628, *p* < 0.001; fear: *r* = 0.549, *p* < 0.001; irritability: *r* = 0.681, *p* < 0.001; disgust: *r* = 0.611, *p* < 0.001; guilt: *r* = 0.416, *p* < 0.001; shame: *r* = 0.461, *p* < 0.001). Finally, two Visual Analogue Scales were administrated to evaluate the adoption of cognitive reappraisal (“I’m trying to see the situation in a different way and in a different light”) and rumination (“I’m continuously rehashing and reflecting on my mood, my emotions, and my reaction to the situation”). Each strategy was rated on a scale from 0 to 100, with 0 indicating no use of the strategy and 100 indicating maximum adoption of the strategy. The ad hoc single items were created after conducting a focus group with four experts in the field of ER. Disagreements were resolved through consensus.

#### 2.2.2. One-Week and One-Month Follow-Up Assessments 

Participants’ autobiographical memories were assessed one week and one month after the occurrence of the event. At the beginning of the study, participants were not told about the autobiographical recall tasks, and they were not aware of the real aim of the study. Whereas the one-week autobiographical recall was conducted in the laboratory by means of a laptop, the one-month follow-up was performed by sending a survey by email (for more details, see Section 2.3 Procedure). 

Both follow-up assessments had the same structure. To recall the negative event recorded in the web link, participants were given the following instructions: “We would like you to recall the event that occurred last week/last month and that you recorded at the web link. You should try to describe and write down the event as it is currently in your mind. Do not worry about the description you gave at the weblink last week/last month: Just describe how much you remember about the episode right at this moment.” Participants were asked to rate the negativity of the episode retrospectively using the same VAS scale as for the momentary assessment (“On a scale from 0 (not unpleasant) to 100 (completely unpleasant), how would you rate the valence of event?”) and to complete the Spanish adaptation [59] of the Memory Characteristic Questionnaire (MCQ) [53], which explores the phenomenological characteristics of a retrieved memory. The MCQ provides five sub-scores: (1) Clarity, which refers to the recall of visual and spatial details of the event; (2) Sensory information, which includes sensory details related to the memory, except the visual ones; (3) Temporal information, which refers to the recall of temporal details about the memory; (4) Emotional involvement, which refers to aspects related to the intensity of feelings, the negative valence of the memory, and the implications/consequences of the event; (5) Thoughts and feelings, which refers to the recall of qualitative features of thoughts and feelings from the episode as well as memories before and after the event. All subscales demonstrated good internal consistency both at one-week (clarity: α = 0.87; sensory information: α = 0.62; temporal information: α = 0.62; emotional involvement: α = 0.79; thoughts and feelings: α = 0.73) and one-month assessments (clarity: α = 0.90; sensory information: α = 0.69; temporal information: α = 0.68; emotional involvement: α = 0.78; thoughts and feelings: α = 0.68). 

### 2.3. Procedure

Participants were undergraduate students recruited through onsite advertisements at the Jaume I University (Castellon, Spain). Potential participants were invited to visit the laboratory to receive more details about the experiment. Individuals who agreed to participate in the study were asked to sign the informed consent and provide demographic information. During this first meeting, the web link to record the negative event was sent via email to the participants. Clear instructions about the experimental task were provided by a researcher from the team. To ensure they understood the procedures, the same instructions were also included in the web link. Participation in the study included a remuneration of 5 euros. Participants were told that to obtain the economic compensation, they had to return to the laboratory one week after reporting their event to complete a final questionnaire.

After reporting the negative event, participants were immediately contacted by a researcher who arranged a face-to-face meeting the following week to provide monetary compensation (approximately 7 days after the occurrence of the event). During this meeting, the participants performed the first “surprise” recall task, which was conducted with a laptop. Of the total initial sample (*n* = 71), 70 participants completed the one-week assessment.

One month after the occurrence of the negative event, an email was sent to the participants with a link to a web survey to complete the second “surprise” memory follow-up. Three reminders (one per day for three consecutive days) were sent to the participants who did not reply to the email. After the third reminder, the assessment was considered missing. The one-month follow-up was completely voluntary, i.e., no further remuneration was given. Of the total sample (*n* = 71), 64 participants completed the one-month follow-up. 

### 2.4. Data Analysis

The dataset of the analyses is contained in an open-access file available in OSF at https://doi.org/10.17605/OSF.IO/N63JZ (accessed on 22 January 2021).

Before analyzing data, we calculated ‘appraisal scores’, which refer to how close the momentary and the retrospective ratings of the event were (i.e., event negativity rated at the time of the initial experience and at the time of the recall). One-week appraisal scores were obtained by calculating the delta values between the one-week ratings and the momentary ratings (i.e., event negativity ratings provided the day of the event). Similarly, one-month appraisal scores were obtained by computing the delta values between one-month and momentary ratings. Positive values indicated the tendency to retrospectively overestimate the negativity of the event, whereas negative scores reflected the tendency to retrospectively underestimate it.

First, correlation analyses were carried out to explore the association between the emotional intensity of the events and the characteristics of the associated memories. Second, the role of ER on autobiographical memory was investigated. To do so, we analyzed the association between cognitive reappraisal, rumination, and autobiographical memory variables. Finally, moderation analyses were conducted to test whether the association between emotional intensity and memory qualities differed as a function of ER. Accordingly, multivariate linear regressions were performed introducing each memory variable (clarity, sensory details, temporal details, emotional involvement, thoughts and feelings, and event appraisal) as the dependent variable for both follow-up assessments separately. The emotional intensity of the event, the two ER strategies (cognitive reappraisal and rumination), and their interaction were the predictors in every model. Considering the consistent body of studies suggesting the key role of the emotional experience in the recall of the associated memory, emotional intensity was introduced as the main regressor in the first block. ER strategies were added in a second block, and the interaction term between strategy and emotional intensity was included in a third block to explore how much this added to the contribution of the predictors prior to the moderation. All predictors were centered before the analyses. The contribution of each ER strategy and each interaction was explored in a separate model to facilitate the interpretation of the results and to reduce multicollinearity problems associated with the inclusion of both cognitive reappraisal and rumination in the same regression. Although a large number of tests were conducted, an alpha level of 0.05 was adopted. The use of more restrictive alpha levels, for instance using a Bonferroni-Holm correction, is less suitable in exploratory studies, because it increases the risk of false negatives by attempting to reduce the risk of false positives [60]. Post hoc calculations of the achieved power were conducted using G*Power (‘linear multiple regression: fixed model, R^2^ increase’) considering a significance level of 5%, a sample size of 70 participants for the one-week follow-up and of 64 participants for the one-month assessment, and the effect sizes obtained with the moderation analyses.

## 3. Results

### 3.1. Descriptive Statistics

In total, 71 negative events were reported (emotional intensity: M = 17.75, SD = 5.06). Event negativity scores were all above 50 (M = 89.01, SD = 12.71), thus confirming their negative valence. Different types of episodes were reported: Argument with a friend (*n* = 19) or with a member of the family (*n* = 10), lovers’ quarrel (*n* = 13), health-related problems (*n* = 6), workplace issues (*n* = 6), object loss or damage (*n* = 10), or other unexpected events, such as failing an exam, missing the train, or having an argument with a stranger (*n* = 7). In all, 30 of the 71 participants (42.3%) adopted cognitive reappraisal (M = 51.63, SD = 27.28) more than rumination (M = 56.87, SD = 28.99). 

On average, participants reported the episode with 402.5 min of delay (SD = 275.83). Nevertheless, the time delay was not significantly associated with neither event negativity (*r* = 0.116, *p* = 0.333) nor emotion intensity (*r* = 0.007, *p* = 0.956), thus suggesting similar episode qualities regardless of the delay. Furthermore, the time delay did not significantly correlate with autobiographical memory qualities, neither at one-week (clarity: *r* = −0.104, *p* = 0.390; sensory information: *r* = 0.153, *p* = 0.207; temporal information: *r* = −0.01, *p* = 0.138; emotional involvement: *r* = 0.141, *p* = 0.244; thoughts and feelings: *r* = 0.068, *p* = 0.577; appraisal: *r* = 0.026, *p* = 0.830) nor at one-month follow-ups (clarity: *r* = 0.033, *p* = 0.798; sensory information: *r* = 0.231, *p* = 0.066; temporal information: *r* = 0.062, *p* = 0.628; emotional involvement: *r* = 0.196, *p* = 0.121; thoughts and feelings: *r* = 0.203, *p* = 0.108; appraisal: *r* = −0.144, *p* = 0.252).

### 3.2. Memory and Emotional Intensity

70 participants completed the one-week follow-up, whereas 64 participants completed the one-month assessment. Significant associations between emotional intensity and autobiographical memory measures are depicted in Table 1.

At one-week and one-month follow-ups, higher emotional intensity was associated with enhanced emotional involvement. Moreover, episodes with higher emotional intensity were recalled with more clarity at the one-month post-assessment, whereas a close-to-significant trend was observed at one-week follow-up (*r* = 0.225, *p* = 0.06). Together, our results confirm the previous literature and suggest that events with higher emotional intensity were more likely to be retrieved with higher clarity and emotional involvement.

### 3.3. Memory and Emotion Regulation: Direct Associations and Moderation Effects

As reported in Table 1, we also analyzed the association between ER and memory characteristics. Rumination was positively associated with emotional involvement both at one-week and one-month follow-ups, as well as with memory thoughts and feelings at a one-month follow-up. Additionally, the use of cognitive reappraisal was associated with remembering more sensory details one week after the occurrence of the episode. 

We, therefore, explored whether ER moderated the association between emotional intensity and autobiographical memory. As Table 2 shows, the inclusion of the interaction term significantly improved the amount of variance explained by emotional intensity and ER separately in some of the models.

First, rumination significantly moderated the relationship between emotional intensity and sensory information both at one-week and one-month follow-ups (Figure 1).

More specifically, low ruminators could retrieve more sensory information as the emotional intensity of the episode increased. The opposite occurred with high ruminators, who were less capable to retrieve sensory information as the emotional intensity of the event increased.

Rumination also significantly moderated the relationship between emotional intensity and emotional involvement both at one-week and one-month follow-ups (Figure 2). More specifically, the emotional involvement with the event increased with its emotional intensity only for low ruminators. At moderate and high levels of rumination, the emotional involvement was relatively high and no longer associated with the intensity of the event, and enhanced emotional involvement was observed also in relation to events with low emotional intensity.

Finally, rumination was a significant moderator of the association between emotional intensity and memory clarity at the one-month assessment (Figure 3). Similar to the previous findings, low ruminators reported remembering more visual and spatial details (i.e., clarity) as the emotional intensity of the event increased. However, this association was less strong in participants adopting medium to high rates of rumination. In fact, individuals with average levels of rumination reported similar levels of clarity regardless of the emotional intensity of the event.

In relation to cognitive reappraisal, only one significant interaction effect was observed. This occurred in the association between emotional intensity and event appraisal at one-month follow-up (Figure 4). Subjects who adopted low to medium levels of cognitive reappraisal tended to overestimate more the negative event as the emotional intensity increased. Unlikely, this association changed its direction in the case of high rates of appraisal, thus showing a greater tendency to underestimate the negativity of the event when the emotional intensity of the episode increased.

## 4. Discussion

The present study aimed to explore the role of ER in the recall of negative autobiographical memories through an ecological approach. Unlike the previous literature, we attempted to assess daily negative events in daily life through an electronic diary and to assess the momentary use of ER strategies to deal with the ongoing negative emotions.

First, we attempted to replicate the findings of the previous research regarding the association between emotions and autobiographical memory [16,17,18]. Our results showed that emotional intensity was associated with the qualities of the recalled autobiographical memories both in the short and long term. More specifically, episodes with higher emotional intensity were associated with enhanced clarity and increased emotional involvement, thus confirming the importance of the emotional experience on the quality of the associated autobiographical memory. 

Second, we explored the association between ER and autobiographical memory characteristics. Overall, our results suggest that ER is associated with and moderates the association between emotional intensity and autobiographical memory and that the use of different strategies entails different outcomes on memory. In the following paragraphs, we will separately discuss the findings of the present investigation in more detail.

### 4.1. Rumination

Rumination refers to the process of persistently thinking about one’s feelings, causes, and consequences. This strategy can be considered maladaptive as it maintains and prolongs negative emotions over time [30,61] and hinders the adoption of problem-solving skills when facing a negative event [28]. Unlike cognitive reappraisal, ruminative reasoning does not promote a constructive change in the meaning of a situation, and it fails to encourage active efforts to cope with it [62], thus leading to negative and passive interpretations of the experience [31]. Moreover, the tendency to ruminate is associated with the selective and enhanced recall of negative stimuli [63]. Hence, it seems plausible that the intense use of rumination in response to a negative event may entail maladaptive outcomes on the subsequent memory.

In the present study, high rumination correlated with enhanced memory of thoughts and feelings at one-month assessment, and with higher emotional involvement at both post-assessments. This increased self-reported memory for cognitive and emotional details could be due to the continuous mental re-creation of the negative situation, which is typical of ruminative thinking [31]. Adding up to the existing literature, rumination was found to moderate the association between the emotional intensity of the event and the associated autobiographical memory.

First, increased emotional intensity was associated with higher memory of sensory details in participants who adopted low rates of rumination (i.e., the higher the emotional intensity, the higher the subjective feeling of remembering details). This was not the case for those who frequently used this strategy. Similar results were observed in relation to clarity (i.e., spatial and visual details) one month after the occurrence of the event, showing a stronger positive association between the two variables only in participants who adopted low rates of rumination. In other words, although the greater emotional intensity is likely to make an event more salient and improve memory performance [11,16], this mechanism seems to be disrupted when using intense rumination. We suggest that the intense use of rumination in response to a negative event may involve a considerable cognitive cost due to an increased focus on the negatively valenced cognitive and emotional components of the experience, which in turn could decrease the attentional resources available to encode and subsequently remember secondary information, such as sensory, visual and spatial details. In this direction, a previous study suggested that rumination may lead to an overloading of the working memory because of a deficit in shutting down negative material, which in turn might result in decreased specific memories [40]. This would explain why, for intense ruminators, memory retrieval was better when the emotional intensity of the event was low. This hypothesis seems to be coherent with an array of studies showing the presence of ‘overgeneral memories’ in depressed individuals (i.e., memories that lack details) [41,42,43], who happen to be characterized by an intense and habitual use of ruminative thinking [31,64]. 

Besides, rumination also significantly moderated the association between the emotional intensity of the reported event and the emotional involvement with the recalled memory both at one week and month follow-ups. According to the results, the more emotionally intense the event was, the higher the emotional involvement with the associated memory became. However, this association was less strong in participants adopting medium-to-high levels of rumination, who showed enhanced emotional involvement also in relation to events with low emotional intensity. These findings suggest that intense ruminators experienced similar levels of emotional vividness, regardless of the emotional intensity of the event, which might be the result of the repetitive thoughts about the episode and the associated negative emotional states [65]. Therefore, our results support the idea that rumination may be considered a maladaptive strategy, as relatively irrelevant experiences seem to require similar processing to emotionally relevant situations. Although this study only assessed rumination in relation to a single negative event, it is possible to hypothesize that intense and regular use of this strategy may have disruptive consequences for memory.

### 4.2. Cognitive Reappraisal

Cognitive reappraisal refers to the attempt to change the interpretation of a stimulus to modify its emotional impact [21]. To date, reappraisal is generally considered an adaptive strategy, that makes it possible to positively re-frame an emotional stimulus [66] and to effectively reduce the intensity of the associated negative emotions [32,67]. Contrary to the typical passive interpretations of ruminative thinking, cognitive reappraisal implies an active role of the individual, who explicitly tries to modify an emotional state by changing the meaning of a situation [26]. As a result of this process, reappraisal may affect the memory associated with a specific negative event, especially from a cognitive point of view.

As hypothesized, cognitive reappraisal was associated with more detailed memories at one-week assessment and, more specifically, with an enhanced recall of the associated sensory information. Even though the recall of details was assessed at a subjective level only, these findings seem to confirm the previous laboratory-based literature, where better memory performance was observed in individuals who adopted reappraisal [33,35]. As Richards et al. (2003) suggested, the attempt to reappraise a stimulus requires a deep conceptual analysis of the situation and an increased focus of one’s attentional resources on its details, which in turn might boost memory performance. Adding up to these previous laboratory findings, our results suggest that the effects of cognitive reappraisal on memory performance may not last over time, thus being only limited to the short term (i.e., one week after the event). Indeed, it is possible that the pass of time, which presumably reduces the emotional impact of an event, makes cognitive reappraisal (and, in turn, remembering the details of the event) no longer necessary.

In our study, cognitive reappraisal was not significantly associated with event appraisal (overestimation or underestimation of the negativity of the event). Interestingly, though, the moderation analyses revealed a significant moderation effect of this strategy in the association between emotional intensity and event appraisal at one-month follow-up. Higher retrospective overestimation of the negativity of the event was observed as the emotional intensity of the episode increased in participants who scarcely adopted reappraisal. This is in line with the evidence showing that people typically tend to retrospectively overestimate negative emotional experiences [57]. This association, however, changed its direction in the case of high rates of appraisal, thus showing a greater tendency to underestimate the negativity of the event when the emotional intensity of the episode was greater. In other words, extensive use of reappraisal seems to be beneficial to reappraise emotionally intense episodes rather than less intense ones, which makes cognitive reappraisal a sensible strategy [68]. Cognitive reappraisal implies a reconstruction and re-elaboration of a stimulus [26], which in turn may lead to a change in one’s subjective estimation of its negativity. Indeed, a less pessimistic representation of a personal negative event may promote an adaptive process of distancing from it, which in turn might reduce its emotional impact over time [21]. Thus, cognitive reappraisal appears to effectively serve its role in modifying the emotional impact of an eliciting stimulus, especially when this is highly intense. Interestingly, these results were only observed at the one-month follow-up, which suggests that the process of reappraising a negative personal event may be cognitively demanding and require time [26].

## 5. Conclusions

Autobiographical memories are an essential part of an individual’s life narrative, allowing one to coherently reconstruct the past, interpret the present, and anticipate the future in terms of an evolving and coherent Self [52]. Although all individuals experience negative events during their lives, the affective and cognitive consequences of these events may differ greatly between individuals. Thus, understanding the factors that influence the encoding and subsequent recall of past memories may be of great importance.

Our results showed that the way people regulate their emotions when facing a negative event may affect their subsequent autobiographical memory. On the one hand, cognitive reappraisal may encourage a deep conceptual analysis of a situation, thus leading to memories that are perceived as richer in detail. Furthermore, the intense use of this strategy may promote an adaptive process of distancing from a negative personal event, which might reduce its emotional impact over time. On the other hand, the putatively maladaptive role of intense rumination emerged, which was associated with the recall of more general memories as the emotional intensity of the event increased, as well as with higher emotional involvement with the episode over time, regardless of the emotional intensity. From an evolutionary point of view, the encoding and recall of negative events represent adaptive processes that guarantee the survival of individuals. In this sense, research has shown that information with higher negative valence is discriminated faster and recalled with more details [16,17], which would reflect an evolutionary mechanism that increases the salience of potentially threatening events [56]. In this sense, rumination might be considered a maladaptive strategy that disrupts these mechanisms by making negative events less detailed in memory. It is important to note that, in our study, we only explored the *momentary* use of ER in response to a specific negative event, which does not necessarily represent the typical ER choice for that person when dealing with other daily episodes [22]. In other words, the findings observed in this study should be interpreted as event-specific. Nevertheless, we might hypothesize that an intense and habitual deployment of reappraisal or rumination in response to negative events may essentially lead to similar adaptive or maladaptive outcomes on the associated memories. 

From a clinical point of view, the findings of the present study have important implications. Patients suffering from depression, for instance, have been shown to be characterized by impaired ER skills and, more specifically, by a reduced use of cognitive reappraisal and enhanced ruminative thinking [24]. Furthermore, impaired autobiographical memory has been observed in depression [69], including a preference to retrieve more negative memories compared to positive ones [70,71], a propensity to overestimate the negativity of past experiences [57], and a tendency to recall memories that lack in details [72,73,74]. Our study supports the relationship between impaired ER and autographical memory deficits and suggests that promoting the use of adaptive strategies (e.g., cognitive reappraisal) and discouraging the adoption of maladaptive ones (i.e., rumination) may also be beneficial for memory. In this direction, reducing ruminative thinking has been found to result in less overgeneral memories [39], which further supports the potential interconnection between ER and autobiographical memory. 

Besides, we acknowledge several limitations of this study. First, the strategies were not assessed with a standardized questionnaire, but rather with ad hoc single items. Although the use of ad hoc items to assess ER is common in EMA studies [61,75,76,77], it is possible that the use of single items did not completely capture the complexity of these strategies. Second, we did not compare the details of the narrations of the events between the different time points. Conversely, memory details were assessed with the MCQ, which measures the subjective feeling of remembering spatial (e.g., “Relative spatial arrangement of objects in my memory for the event is: 1 = vague; 7 = clear/distinct”), visual (e.g., “My memory for this event is: 1 = sketchy; 7 = very detailed”), sensory (e.g., “My memory for this event involves sounds: 1 = little or none; 7 = a lot”) and temporal details (e.g., “My memory for the day when the event takes place is: 1 = vague; 7 = clear/distinct”). Therefore, the indexes used to assess memory details had a self-reported nature. Moreover, the MCQ was originally designed to assess and differentiate memories of external (i.e., experienced) and internal (i.e., imagined) origin [53]. Thus, its use might have not been fully appropriate to assess the research questions of the present research. Third, the sample was mainly of female sex and composed of undergraduate students. Considering the evidence showing potential age [78,79] and gender differences in ER [80,81], as well as the phenomenological differences of autobiographical memories across the adult lifespan [82], a replication of this study is needed to confirm the results observed. Similar studies that investigate the deployment of a broader range of ER strategies are also recommended. Fourth, future experiments should also be conducted to explore the causal rather than correlational nature of the results reported in this study. For example, random prompts could be sent to participants throughout the day to ask whether a negative event is currently occurring. At times, participants could be instructed to regulate their emotions using a specific strategy, whereas at other times they would not receive any instruction. This study design would make it possible to compare the recall of negative life episodes that have been manipulated with those that have not and, thus, draw causal conclusions. Fifth, adopting an ecological approach involves less control over confounding variables, so our design may have stronger ecological validity at the expense of internal validity [83]. Accordingly, when experiencing a negative episode, individuals did not remember to immediately use the link and write down their memories, as revealed by the presented time delays in reports. Nevertheless, we tried to control this by deleting events reported after over 12 h of delay. Although no studies are showing to what extent memory accuracy can change within a few hours, it is possible that the time delay affected our results. Importantly, though, no significant differences in the quality of the events and autobiographical memories were observed depending on the time delay. Additionally, participants were not prompted to report an event; instead, they had to spontaneously enter the web link. Thus, the selection of the negative event may itself be a confounding factor, and the selected episode may be qualitatively different from other negative events that occurred in participants’ daily lives but were not written down. Also, the act of reporting the event on the web link implied an additional encoding of the episode, which in turn could have affected the quality of the memories. Finally, it is important to note that some of the moderation analyses turned out to be underpowered. While acknowledging this, it is important to note that almost all main study analyses were sufficiently powered. In addition, the findings of the present study in terms of effect sizes and power analyses might help researchers to calculate the necessary sample size in future similar studies.

## Figures and Tables

**Figure 1 ijerph-18-07122-f001:**
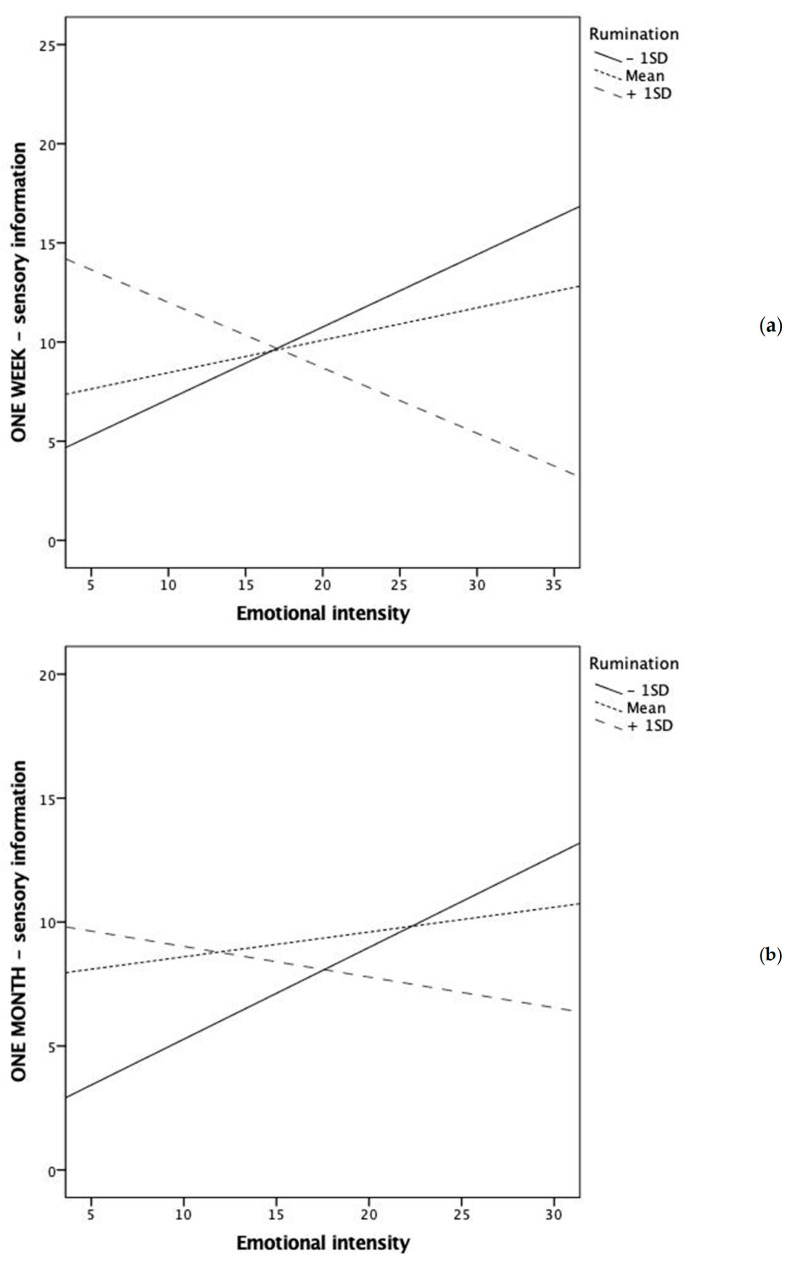
Graphical representation of the significant interaction between rumination and emotional intensity in predicting emotional involvement at one-week (**a**) and one-month (**b**) follow-up assessments.

**Figure 2 ijerph-18-07122-f002:**
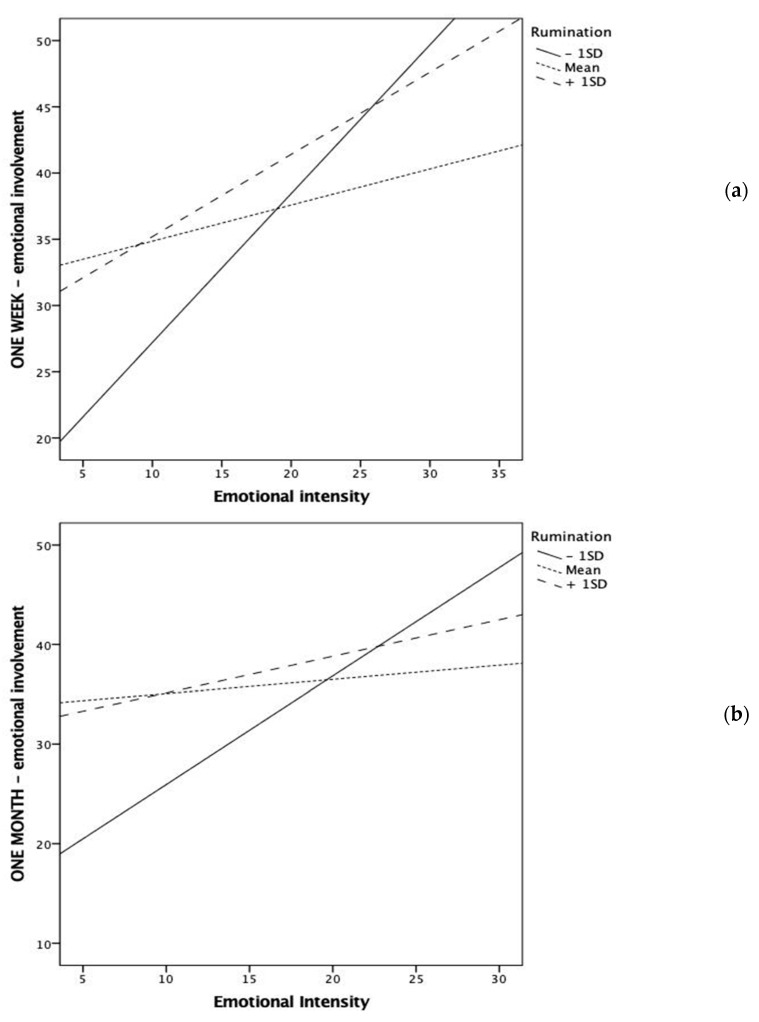
Graphical representation of the significant interaction between rumination and emotional intensity in predicting sensory information at one-week (**a**) and one-month (**b**) follow-up assessments.

**Figure 3 ijerph-18-07122-f003:**
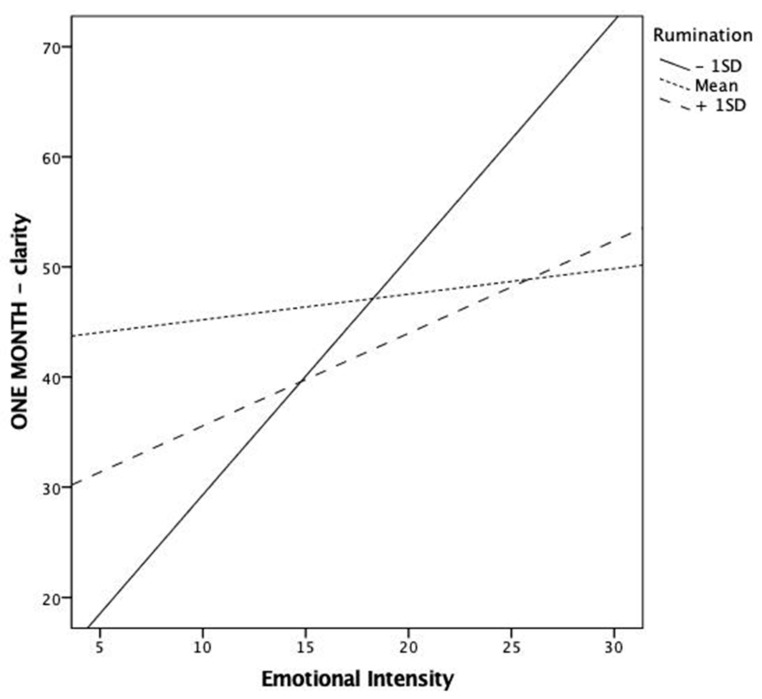
Graphical representation of the significant interaction between rumination and emotional intensity in predicting clarity at a one-month follow-up.

**Figure 4 ijerph-18-07122-f004:**
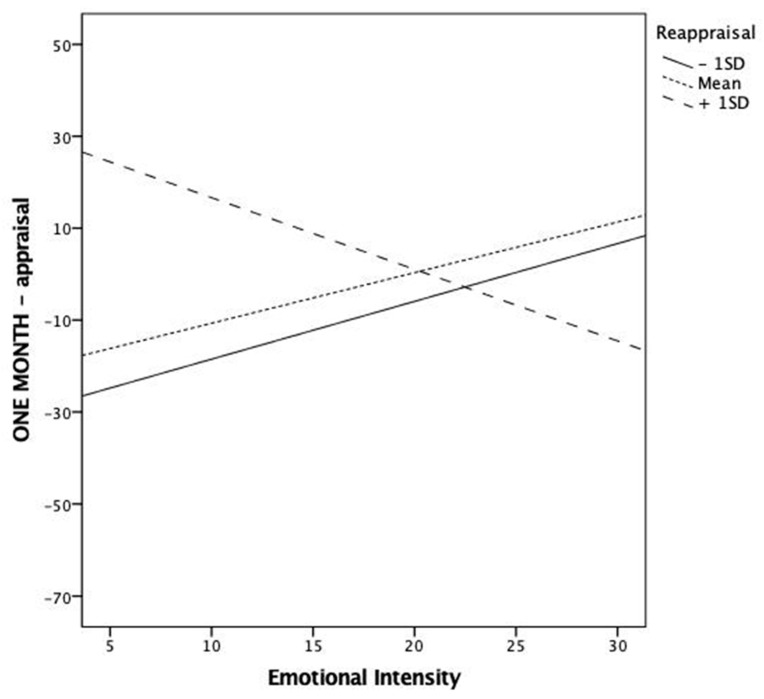
Graphical representation of the significant interaction between cognitive reappraisal and emotional intensity in predicting event appraisal at a one-month follow-up. Note that positive appraisal scores reflect the retrospective overestimation of the negativity of the event, whereas negative values indicate the tendency to retrospectively underestimate it.

**Table 1 ijerph-18-07122-t001:** Correlations of emotional intensity and emotion regulation with autobiographical memory characteristics were assessed through the Memory Characteristics Questionnaire (MCQ). Change in appraisal scores were obtained by calculating the absolute delta scores between the momentary ratings and the one-week/one-month follow-up ratings.

One Week (*n* = 70)			
	Emotional intensity	Rumination	Reappraisal
MCQ-Clarity	0.225	−0.025	0.056
MCQ-Sensory information	0.048	−0.039	0.297 *
MCQ-Temporal information	0.179	0.113	0.133
MCQ-Emotional involvement	0.428 ***	0.244 *	−0.008
MCQ-Thoughts and feelings	0.223	0.190	0.103
Appraisal	−0.066	0.180	0.081
**One Month (*n* = 64)**			
	Emotional intensity	Rumination	Reappraisal
MCQ-Clarity	0.284 *	0.108	−0.121
MCQ-Sensory information	0.146	0.079	0.043
MCQ-Temporal information	0.108	0.164	0.079
MCQ-Emotional involvement	0.312 **	0.308 *	−0.075
MCQ-Thoughts and feelings	0.151	0.293 *	0.122
Appraisal	0.006	0.175	0.202

* *p* < 0.05, ** *p* < 0.01, *** *p* < 0.001.

**Table 2 ijerph-18-07122-t002:** Multivariate linear regressions predicting autobiographical memory characteristics. The Betas are from the last step of the regression equation. All predictors were centered. Post-hoc achieved power was calculated using G*Power.

ONE WEEK RECALL (*n* = 70)																		
	MCQ-Clarity			MCQ-Sensory Information			MCQ-Temporal Information			MCQ-Emotional Involvement			MCQ-Thoughts & Feelings			Event Appraisal		
Predictors	β [95% CI]	ΔR^2^	Achieved power	β [95% CI]	ΔR^2^	Achieved power	β [95% CI]	ΔR^2^	Achieved power	β [95% CI]	ΔR^2^	Achieved power	β [95% CI]	ΔR^2^	Achieved power	β [95% CI]	ΔR^2^	Achieved power
Emotional intensity	0.172	0.038		0.137	0.034		0.191	0.048		0.328	0.184 ***		0.162	0.055		0.126	0.021	
Rumination	−0.046	0.002		−0.070	0.004		0.034	0.001		0.200	0.041		0.134	0.018		0.173	0.027	
Interaction	−0.177 [−0.028; 0.004]	0.030	0.619	−0.338 [−0.017; −0.003]	0.110 **	0.929	−0.102 [−0.010; 0.004]	0.010	0.542	−0.277 [−0.024; −0.003]	0.074 **	0.999	−0.207 [−0.017; 0.001]	0.041	0.841	0.116 [−0.014; 0.041]	0.013	0.500
Emotional intensity	0.196	0.038		0.192	0.034		0.222	0.048		0.428	0.184 ***		0.236	0.055 *		0.145	0.021	
Reappraisal	0.020	0.000		0.216	0.046		0.113	0.013		−0.034	0.001		0.073	0.005		0.004	0.000	
Interaction	0.092 [−0.012; 0.026]	0.008	0.447	0.056 [−0.006; 0.010]	0.003	0.698	−0.084 [−0.011; 0.005]	0.007	0.607	0.088 [−0.008; 0.018]	0.008	0.981	0.156 [−0.004; 0.017]	0.024	0.701	−0.017 [−0.034; 0.030]	0.000	0.213
**ONE MONTH RECALL (*n* = 64)**																		
	**MCQ-Clarity**			**MCQ-Sensory Information**			**MCQ-Temporal Information**			**MCQ-Emotional Involvement**			**MCQ-Thoughts & Feelings**			**Event Appraisal**		
Predictors	β [95% CI]	ΔR^2^	Achieved power	β [95% CI]	ΔR^2^	Achieved power	β [95% CI]	ΔR^2^	Achieved power	β [95% CI]	ΔR^2^	Achieved power	β [95% CI]	ΔR^2^	Achieved power	β [95% CI]	ΔR^2^	Achieved power
Emotional intensity	0.329	0.081		0.161	0.021		0.130	0.032		0.266	0.098 *		0.067	0.023		0.109	0.025	
Rumination	−0.074	0.000		−0.008	0.002		0.137	0.018		0.165	0.044		0.251	0.065 *		0.115	0.004	
Interaction	−0.376 [−0.057; −0.013]	0.135 **	0.985	−0.270 [−0.017; −0.001]	0.070 **	0.713	−0.034 [−0.013; 0.010]	0.001	0.446	−0.277 [−0.028; −0.002]	0.073 *	0.984	−0.107 [−0.016; 0.006]	0.011	0.743	0.190 [−0.009; 0.062]	0.034	0.536
Emotional intensity	0.277	0.081 *		0.160	0.021		0.188	0.032		0.316	0.098 *		0.162	0.023		0.150	0.025	
Reappraisal	−0.112	0.012		0.087	0.006		0.070	0.004		−0.056	0.004		0.135	0.017		0.148	0.029	
Interaction	−0.050 [−0.030; 0.020]	0.002	0.724	0.158 [−0.003; 0.014]	0.025	0.454	0.114 [−0.007; 0.018]	0.013	0.428	0.080 [−0.010; 0.019]	0.006	0.780	0.085 [−0.008; 0.016]	0.007	0.416	−0.258 [−0.075; −0.002]	0.066 *	0.835

* *p* < 0.05, ** *p* < 0.01, *** *p* < 0.001.

## Data Availability

The dataset of the analyses is contained in an open-access file available in OSF at https://doi.org/10.17605/OSF.IO/N63JZ.

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
