# Peer review of "The Moderating Role of Emotion Regulation in the Recall of Negative Autobiographical Memories"

_ijerph, 2021, doi:10.3390/ijerph18137122_

Round 1

Reviewer 1 Report

This ms describes an experiment with 71 undergraduate mainly female students who were instructed to grab a negative event with significant impact that occurred spontaneously during the study period. They were asked to self-report a number of variables related to the event. Without forewarning they were approached again after one week and then again after one month and asked to describe the same event and its characteristics in retrospect. Variables studied were emotional intensity, rumination and reappraisal. These variables were related to a number of characteristics of they way in which the experimental persons described the event, clarity, sensory information, temporal information, emotional involvement, thoughts and feelings and finally appraisal.

30 or 71 participants were judged from the questionnaire responses to have used cognitive appraisal rather than rumination. A raw summary of the findings is that subjects who used reappraisal had a more constructive way of coping with the negative event.

The text is well written in good and educational language. The authors have performed power calculations and on the whole handled the statistics professionally.

As the authors rightly point out themselves, there are a number of difficulties that arise when we think of the possibility to generalize. We are dealing with undergraduate students with a clear overrepresentation of women in Spain. Although the selected events seem to represent moderately dramatic negative events in the lives of young students we of course do not know to what extent they represent people in the general population experiencing other kinds of negative events. In addition it is hard to know whether the choice (between rumination and reappraisal) that the participants made themselves would be the same in other situations for those participants. In other words, did the choice have to do with the nature of the event under those special conditions or did the choice represent a typical choice for that person? To what extent is the choice influenced by any emotional state (for instance mild depression) that the person happens be be in for the moment?

Author Response

This ms describes an experiment with 71 undergraduate mainly female students who were instructed to grab a negative event with significant impact that occurred spontaneously during the study period. They were asked to self-report a number of variables related to the event. Without forewarning they were approached again after one week and then again after one month and asked to describe the same event and its characteristics in retrospect. Variables studied were emotional intensity, rumination and reappraisal. These variables were related to a number of characteristics of they way in which the experimental persons described the event, clarity, sensory information, temporal information, emotional involvement, thoughts and feelings and finally appraisal. 30 or 71 participants were judged from the questionnaire responses to have used cognitive appraisal rather than rumination. A raw summary of the findings is that subjects who used reappraisal had a more constructive way of coping with the negative event.

The text is well written in good and educational language. The authors have performed power calculations and on the whole handled the statistics professionally.
As the authors rightly point out themselves, there are a number of difficulties that arise when we think of the possibility to generalize.

We are very pleased to know that the reviewer believes that the manuscript might represent an important contribution. We are also very grateful for the insightful points outlined. We have enthusiastically tried to address all his/her concerns. In our resubmitted manuscript, we have highlighted areas of substantive changes in yellow, so that the Editor and the reviewer can easily identify them.

We are dealing with undergraduate students with a clear overrepresentation of women in Spain. Although the selected events seem to represent moderately dramatic negative events in the lives of young students we of course do not know to what extent they represent people in the general population experiencing other kinds of negative events.

We completely agree with the reviewer about the limitation of the sample. The sample of our study included undergraduate students, which might reduce the possibility to generalize our findings. In this sense, both age (see for example Blanchard Fields et al., 2004; or Orgeta 2009) and sex (see for example Goubet and Chrysikou, 2019; McRae et al., 2008) have been found to shape emotion regulation processes. Furthermore, the phenomenological characteristics of recalled autobiographical memories have been shown to change across the adult lifespan, with higher ratings of phenomenology (e.g., vividness of turning points) observed in older adults compared to younger individuals (Luchetti and Sutin, 2018). We have now underlined these limitations in more detail in the conclusion section, including some references (from line 600).

In addition it is hard to know whether the choice (between rumination and reappraisal) that the participants made themselves would be the same in other situations for those participants. In other words, did the choice have to do with the nature of the event under those special conditions or did the choice represent a typical choice for that person? To what extent is the choice influenced by any emotional state (for instance mild depression) that the person happens to be in for the moment?

This is a very interesting point. Almost all previous research exploring emotion regulation and memory are based on a trait-based conceptualization of emotion regulation, which is considered to be a stable and cross-situational tendency of an individual. Nevertheless, there is evidence showing that ER is more likely to be a situated process shaped by momentary situational and contextual factors. Indeed, we do believe that one of the strengths and novelties of this study is the fact that we did not use a trait-based questionnaire to assess ER: Instead, we explored to what extent participants adopted cognitive reappraisal and rumination in the specific situation reported in the diary. While acknowledging the importance of trait ER, as noted above there is evidence to suggest that strategy deployment is to a significant extent associated with the characteristics of the event and the associated emotional experience, rather than being a cross-situational tendency of a person. For instance, previous studies have revealed a positive correlation between the use of rumination, momentary negative affect and event severity (Li et al., 2017; Moberly & Watkins, 2008), as well as a higher implementation of cognitive reappraisal in less emotionally intense situations (Sheppes et al., 2014).

To sum up, the findings of our study are specific for those events reported in the diary and subsequently recalled at the one-week and one-month follow-ups. However, we might expect that an intense and habitual use of those strategies could have similar effects on memory for those events in which the same pattern of strategies is observed. For instance, an intense and repetitive use of rumination might be maladaptive in terms of autobiographical memory phenomenology, which indeed happens to be disrupted in depressed individuals who are prone to use ruminative thinking frequently. We have discussed this in more detail in the text now (from line 564).

Reviewer 2 Report

This study proposes a moderation model for the relationship of autobiographical memory to emotional intensity, moderated by emotional regulation. 

I would like to congratulate the authors, as I consider it to be a well-structured, original and well-worked paper. I personally enjoyed reading it very much, and I am sure it will be a great contribution to the literature in this field.

The work has great strengths and I can only make minor points of which some are optional for the authors: one methodological and another one in a theorical level:

1. Although analysis are correctly conducted and reported, for transparency, it would be helpful to report the regression coefficient CIs for the interaction terms . But as authors have uploaded the dataset, this is up to the authors.

2. Around lines 554, the role of negative emotional charge is discussed. In this way, much has been said about the role of negative emotions in human survival. This is a reflection that I consider interesting for work and future lines of research. I suggest literature on the subject, which is obviously optional for the authors to include. Particularly because these pieces or work are mostly based on recognition task and as I am co-author of one of the publications (but maybe could be of interest):

Gordillo, F., Arana, J. M., Mestas, L., Salvador, J., Meilán, J. J. G., Carro, J., & Pérez, E. (2010). Emotion and recognition memory: The discrimination of negative information as an adaptive process.

Kugler, L., & Kuhbandner, C. (2015). That’s not funny!–But it should be: effects of humorous emotion regulation on emotional experience and memory. Frontiers in psychology6, 1296.

Moret‐Tatay, C., Moreno‐Cid, A., Argimon, I. I. D. L., Quarti Irigaray, T., Szczerbinski, M., Murphy, M., ... & Fernández de Córdoba Castellá, P. (2014). The effects of age and emotional valence on recognition memory: An ex‐Gaussian components analysis. Scandinavian journal of psychology55(5), 420-426.

PS. Thanks for making this good work part of OSF. This says a lot about the authors' integrity towards possible replications.
Also, after reading the paper, I am not surprised that it is funded by programmes of excellence such as MSCA or conducted by Botella's team. The current work really reflects this high quality standard. My congratulations once again.

Author Response

REVIEWER 2

This study proposes a moderation model for the relationship of autobiographical memory to emotional intensity, moderated by emotional regulation. 
I would like to congratulate the authors, as I consider it to be a well-structured, original and well-worked paper. I personally enjoyed reading it very much, and I am sure it will be a great contribution to the literature in this field.
The work has great strengths and I can only make minor points of which some are optional for the authors: one methodological and another one in a theorical level
.

We are very pleased to know that the reviewer believes that the manuscript might represent an important contribution to the literature. Also, it was a pleasure to read such a nice comment on our manuscript, we do believe that the suggestions received during the review process have increased the quality of our article.

We have enthusiastically tried to address all the reviewer’s concerns. In our resubmitted manuscript, we have highlighted areas of substantive changes in yellow, so that the Editor and the reviewer can easily identify them.

Although analysis are correctly conducted and reported, for transparency, it would be helpful to report the regression coefficient CIs for the interaction terms . But as authors have uploaded the dataset, this is up to the authors.
We have now included the CIs coefficients for the interaction terms in table 2.

Around lines 554, the role of negative emotional charge is discussed. In this way, much has been said about the role of negative emotions in human survival. This is a reflection that I consider interesting for work and future lines of research. I suggest literature on the subject, which is obviously optional for the authors to include. Particularly because these pieces or work are mostly based on recognition task and as I am co-author of one of the publications (but maybe could be of interest)

Thank you very much for the suggestion. We have now included the suggested studies by Grodillo et al. (2015) and Kugler et al (2015) in the manuscript (from line 157; line 553; from line 558).
